# Global patterns of potential future plant diversity hidden in soil seed banks

Xuejun Yang [1], Carol C. Baskin[2,3], Jerry M. Baskin[2], Robin J. Pakeman[4], Zhenying Huang[1✉], Ruiru Gao[5] & Johannes H. C. Cornelissen[6]

Soil seed banks represent a critical but hidden stock for potential future plant diversity on Earth. Here we compiled and analyzed a global dataset consisting of 15,698 records of species diversity and density for soil seed banks in natural plant communities worldwide to quantify their environmental determinants and global patterns. Random forest models showed that absolute latitude was an important predictor for diversity of soil seed banks. Further, climate and soil were the major determinants of seed bank diversity, while net primary productivity and soil characteristics were the main predictors of seed bank density. Moreover, global mapping revealed clear spatial patterns for soil seed banks worldwide; for instance, low densities may render currently species-rich low latitude biomes (such as tropical rain-forests) less resilient to major disturbances. Our assessment provides quantitative evidence of how environmental conditions shape the distribution of soil seed banks, which enables a more accurate prediction of the resilience and vulnerabilities of plant communities and biomes under global changes.

[1] State Key Laboratory of Vegetation and Environmental Change, Institute of Botany, Chinese Academy of Sciences, Beijing, China. [2] Department of Biology, University of Kentucky, Lexington, KY, USA. [3] Department of Plant and Soil Sciences, University of Kentucky, Lexington, KY, USA. [4] Department of Ecological Sciences, The James Hutton Institute, Craigiebuckler, Aberdeen AB15 8QH, UK. [5] The School of Life Sciences, Shanxi Normal University, Linfen, Shanxi, China. [6] Systems Ecology, Department of Ecological Science, VU University, De Boelelaan 1085, 1081 HV Amsterdam, The Netherlands. ✉email: zhenying@ibcas.ac.cn

S oil seed banks are vital for the long-term survival of individual plant species and the diversity and dynamics of plant communities[1]. Thereby, they represent a critical but hidden stock for potential future plant diversity on Earth. Seed banks, which include all viable seeds on or in the soil, vary spatially and temporally[2]. Ecological and evolutionary theory recognizes seed banks as 'biodiversity reservoirs'. Indeed, seed banks support population persistence and biodiversity maintenance through temporal storage effects[3] and increasing the gene pool[4], thereby maintaining a diverse but hidden species pool belowground that hedges against risk of environmental change[3]. Further, seed banks can affect the potential rate and even direction of evolutionary change because they increase the mean generation times of populations[5,6]. Therefore, clarifying the functions of seed banks in community and population dynamics is a key challenge for understanding basic ecological patterns and processes[7].

Patterns and variation of soil seed banks have long been of much popular interest[8]. Despite the extremely heterogeneous nature of soil seed banks[9], most studies have been conducted at the local scale, which hampers our general understanding of the assembly processes of this biodiversity reservoir at large scales[10]. The few recent studies that have reported on patterns of soil seed banks at macroscales have been conducted in certain regions (e.g., Europe[11]) or at the global scale for a specific plant group (e.g., invasive species[12]) or ecosystem (e.g., grasslands[10,13]).

Further, the very low similarity between soil seed banks and the standing vegetation has been widely recognized[6,9,10]; thus, both environmental determinants and responses to global change differ fundamentally between them. Given the predicted impacts of global change on biodiversity, effective management of global diversity requires a complete understanding of the response of plant diversity to environmental changes both aboveground (standing vegetation) and belowground (storage organs, bud banks and soil seed banks). Soil seed bank diversity and density represent much of the resilience of local to biome-scale plant diversity in the face of major disturbances linked to climate or land-use changes. Fully understanding the geographical distribution and environmental determinanats of soil seed banks[14], and modeling their role in future plant diversity requires a global assessment that disentangles the effects of environmental gradients on soil seed bank diversity and density.

Here, we provide such assessment by compiling and analyzing an extensive database to characterize global determinants and patterns of soil seed banks. Close relationships between the soil seed banks and environmental variables (including climate and soil) have been reported, albeit mostly at the local or regional scale[5,9,10]. We hypothesized that soil seed bank composition and density should show clear global patterns since environmental conditions vary geographically across the Earth. Biologically, since seed dormancy and longevity in the soil are determined by temperature, precipitation and soil environments[1] and climate and soil drive plant productivity that in turn should drive seed influx into the soil, we further hypothesized that climate and soil variables are important for predicting soil seed bank diversity and density at the global scale. The main results of our study show that diversity of soil seed banks exhibits clear latitudinal patterns. Climate and soil are the major determinants of seed bank diversity, while net primary productivity and soil characteristics are the main predictors of seed bank density. These results provide insights into environmental determinants of soil seed banks at the global scale.

## Results and discussion
Our global database was derived from studies measuring soil seed bank diversity and density of natural plant communities across all continents, albeit with a strong data availability bias towards North America, Europe, eastern Asia and Oceania as compared to elsewhere (Fig. 1). The database contains 15,698 records for soil seed banks worldwide, including 6,480 for diversity (represented here by species richness) and 9,218 for density (number of seeds per soil surface area). The database represents more than a century of research with the oldest publication dating back to 1918[15]. This most exhaustive and comprehensive set of research data on soil seed bank to date allowed us to identify the determinants and patterns of soil seed bank at the global scale.

To make data among studies comparable, we standardized them using a three-step process. First, we identified soil seed banks that showed seasonal patterns in both diversity and density, all of which peaked slightly in winter (Supplementary Fig. 1a, b). Thus, we standardized all data (from non-(sub-)tropical regions) for other seasons to winter. Second, sampling area for soil seed bank diversity varied among studies, with 0.01 m² being the most commonly reported (Supplementary Fig. 2), to which we standardized all data using a species-area curve (Supplementary Table 1). Third, sampling depth also varied among studies, with 0–5 cm being the most frequently reported soil depth (Supplementary Fig. 3). Therefore, 0–5 cm was chosen as the soil depth for standardization of data in various soil depths. Such standardization is needed to find the relationships of seed bank data between different soil depths. We used the upper and lower limits of soil depths (e.g., for 0–5 cm, the upper limit was 0 cm and the lower 5 cm). The log-scale regressions showed that both soil seed bank diversity and density decreased significantly with lowering upper boundaries of soil depths but increased with lower ones (Supplementary Table 2), and thus we standardized all data to 0–5 cm depth using these relationships. To account for possible variation among biomes, the second and third standardization procedures were conducted for each biome separately. The analyses during standardization confirmed the need to standardize empirical findings when comparing seed bank patterns across studies, as previously stressed in a study on grassland soil seed banks[10]. Our standardization procedures made all data comparable in terms of season, sampling area and soil depth.

Non-parametric Kruskal–Wallis tests showed that soil seed banks differed significantly among ecosystem types. Mangroves, tundra and tropical & subtropical dry broadleaf forests had a lower diversity of soil seed banks, whereas Mediterranean forests, woodlands & scrub, tropical & subtropical moist broadleaf forests and tropical & subtropical coniferous forests had a higher diversity (Supplementary Fig. 4a). For density, mangroves and flooded grasslands & savanna had the lowest value, while temperate broadleaf & mixed forests and temperate conifer forests had the highest value (Supplementary Fig. 4b).

Prior to spatial analyses, we computed semivariograms to determine whether spatial autocorrelation could affect our models. We found that there was no obvious spatial autocorrelation in the data of soil seed bank diversity or density (Supplementary Fig. 5), indicating no spatial dependence in our data. We then used the random-forest algorithm (see Methods for details) to determine the importance (as increase in node purity) of the influence of 31 variables related to climate, soil, human disturbance and spatial coordinates (Supplementary Table 3) on diversity and density of soil seed banks. These variables previously were reported to affect plant performance at the global scale[16–18], and thus they could affect soil seed banks via their effects on seed production. Moreover, we expected that potentially these variables could affect seed longevity in the soil. Full models using all 31 predictors showed that climate and soil were important in predicting soil seed banks (Fig. 2a and Supplementary Fig. 6). Moreover, spatial coordinates (absolute latitude) were the most important predictor for diversity, i.e., diversity of

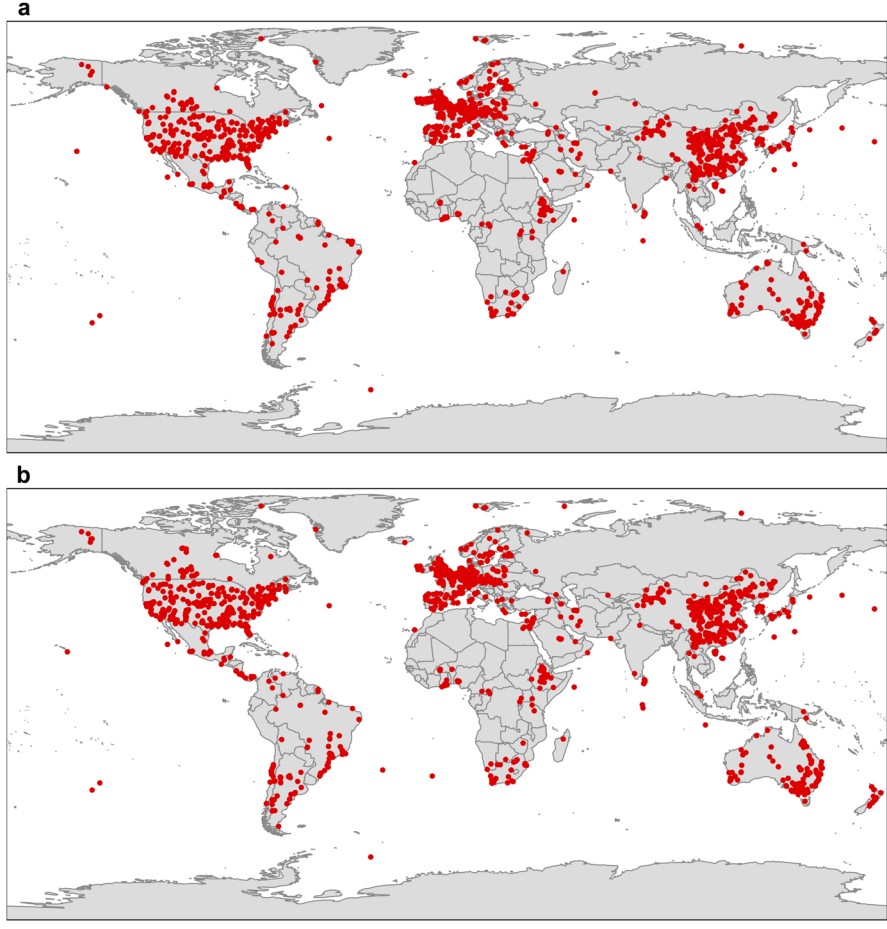

**Fig. 1 Locations of the soil seed bank studies included in our database. a** Diversity; **b** Density.

soil seed banks exhibit clear spatial patterns at the global scale. Net primary productivity (NPP) and soil characteristics were important in predicting the density of soil seed banks (Fig. 2b and Supplementary Fig. 6).

We then built final random-forest models using the most important predictors of seed banks selected from full models: nine variables for diversity and five for density (Supplementary Fig. 7). Final models explained more of the total variance than did full models (Supplementary Table 4), and they were robust to K-fold cross-validation (Supplementary Fig. 8), indicating that a small number of variables predicted soil seed bank diversity and density. Absolute latitude (abs.latit) was the most important predictor for diversity, which varied between 0–55° and then decreased beyond this range (Fig. 3a). Five climatic variables were important for diversity. Diversity peaked at intermediate annual temperature ranges (ATR), while it was the lowest at intermediate mean temperature of driest quarter of the year (TDQ), precipitation of the coldest quarter (PCQ) and precipitation of the driest quarter (PDQ). Diversity increased with increasing annual precipitation (AP). In addition, three soil variables were important for diversity. Diversity showed a humped relationship with soil pH, with pH 6–7 having the highest diversity. Diversity increased with soil cation exchange capacity (CEC) and soil silt content (SILT). These results indicate that diversity exhibits strong spatial patterns at the global scale. However, our spatial patterns differ from those found for a specific ecosystem worldwide (e.g., grasslands), where there were only weak latitudinal gradients in seed bank diversity[10]. In addition, climate emerged as an important predictor for seed bank diversity, which is consistent with the report that climate acts as environmental filters

affecting soil seed bank of grasslands around the world[13]. Our results agree with a continental study in Europe, where ATR was more important than mean annual temperature for determining seed bank richness and warmer temperatures were associated with lower seed bank richness[11]. Possible mechanisms by which temperature affects soil seed banks are that it (1) influences seed bank inputs via its effects on seed production; (2) cues dormancy-breaking and germination[1], thus determining germinable seed output from seed banks; and (3) affects seed metabolic activity and soil fungal activity[19], thereby determining seed viability and persistence in the soil. Finally, our findings of a significant effect of soil pH are supported by some regional and local studies. For instance, seed bank composition is significantly associated with soil pH at high elevations on the Tibetan Plateau[20]. A negative effect of low pH also has been reported in a large-scale study of acidic and calcareous grasslands in England[21]. Two possible mechanisms for the effects of soil pH are that (1) low pH may cause loss of seed viability due to the toxicity from aluminum or other metals that become more readily available in soils with low pH[22]; and (2) high pH may accelerate decomposition and promote growth of pathogens that negatively affect seed persistence[23]. In our study, the two mechanism may operate synchronously, thereby resulting in the highest diversity of soil seed banks at intermediate pH at the global scale. Further, our results show that soil CEC and SILT affect seed bank diversity, which agrees with a study on the Tibetan Plateau[20]. The physical and chemical properties of soils can affect seed bank directly by affecting seed germination and aging via regulating soil water-holding capacity[24], or indirectly by affecting seed viability via controlling the activity of soil pathogens[21,22,25].

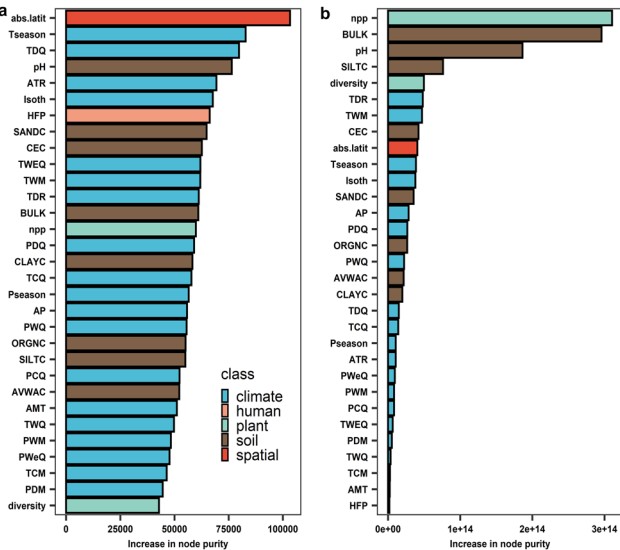

**Fig. 2 Variable importance (increase in node purity) of random forests run with all 31 predictors. a** Soil seed bank diversity; **b** Density. abs.latit, absolute latitude; AMT, annual mean temperature; AP, annual precipitation; ATR, annual temperature range; AVWAC, available water capacity (%); BULK, bulk density; CEC, cation exchange capacity; CLAYC, clay (mass %); diversity, plant diversity; HFP, human footprint; Isoth, isothermality; npp, plant productivity (net primary production); ORGNC, organic carbon content; PCQ, precipitation of coldest quarter; PDM, precipitation of driest month; PDQ, precipitation of driest quarter; pH, pH measured in water; Pseason, precipitation seasonality (coefficient of variation); PWeQ, precipitation of wettest quarter; PWM, precipitation of wettest month; PWQ, precipitation of warmest quarter; SANDC, sand (mass %); SILTC, silt (mass %); TCM, min temperature of coldest month; TCQ, mean temperature of coldest quarter; TDQ, mean temperature of driest quarter; TDR, mean diurnal range (mean of monthly (max temp–min temp)); Tseason, temperature seasonality (standard deviation *100); TWeQ, mean temperature of wettest quarter; TWM, max temperature of warmest month; TWQ, mean temperature of warmest quarter.

For soil seed bank density, soil bulk density (BULK) was the most important predictor; density increased below 750 g/cm$^3$ BULK but remained stable when BULK was higher than 800 g/cm$^3$ (Fig. 3b). Density peaked when temperature of the warmest month (TWM) was 34 °C. Density showed similar variation with NPP, precipitation of the driest quarter of the year (PDQ) and of the driest month (PDM), i.e., it peaked at intermediate values of these variables. Precipitation influences the success of sexual reproduction of plants and the size of the seed bank through seed input[26], and it also affects soil pathogenic fungi, which cause seed mortality[27]. Therefore, precipitation has a strong effect on seed bank density, as reported for 27 alpine meadows on the Tibetan Plateau[28]. Our results further illustrate that PDQ and PDM are the key factors determining seed bank density worldwide, suggesting that moisture fluctuation in soils triggered by precipitation of the driest time of the year can affect seed bank density. If soil moisture fluctuations are high, seed germination will be primed by increasing moisture[24].

At the global scale, we mapped soil seed bank diversity and density using the final random-forest models. Mapping soil seed bank values onto global maps revealed considerable geospatial variation, the pattern of which varied between diversity and density (Fig. 4). For diversity, western North America, central South America, central Africa, central Europe, southern and eastern Asia and eastern Oceania had high values. In contrast, eastern and central North America, northern Africa and central

Asia had low values (Fig. 4a). For density, northern North America, northern Europe and northern Asia had higher values than elsewhere (Fig. 4a). Our results are consistent with the reports that larger seed banks are more common in cooler temperate climates[19,29]. The latitudinal pattern of higher density in colder regions in the Northern Hemisphere may be driven by lower seed mortality in colder soils[6], resulting in stable seed bank densities of long-lived seeds that counteract low seed production in some years at cold northern latitudes, as shown in a study of temperate forests along a 1900 km latitudinal gradient in northwestern Europe[29]. The latitudinal pattern highlights that particularly species rich low-latitude biomes such as tropical rainforests generally have very low seed bank densities, while their seed bank diversity does not exceed that in higher latitudes biomes. However, our global assessment should be interpreted with caution since some studies in azonal vegetation or in rare habitats in our database did not fully reflect soil seed banks in that region, and thus these data shortcomings may have induced bias in our global predictions. Moreover, data gaps in our database are also likely to have had an effect on the global predictions, i.e., fewer data available from some continents (e.g., northern Asia and Africa) could lead to less confidence for prediction in these regions. For example, Russia has very few soil seed bank data, which may have led to an inaccurate prediction for this country. Nevertheless, based on our global patterns of soil seed bank diversity and density, the latitudinal pattern strongly suggests that the biodiversity of (sub-)tropical forests is particularly vulnerable to large-scale climatic or land-use disturbances. However, in-depth investigation is needed to quantify the extent to which temporal integration of seed bank effects for long-lived trees and seed masting events may buffer the effects of low seed bank diversity and density at any given time of sampling. In contrast, the higher-latitude plant diversity, while currently low compared to that in tropical rainforest, may rely on high soil seed bank densities to boost its resilience to large-scale climate- or land-use induced disturbances. Further, our analyses suggest that the least vulnerable ecosystems in terms of hidden diversity should be those that combine high seed-bank diversity with high density; and therefore the relationships between the two variables across the global map certainly would be an interesting topic worthy of further study.

Our global assessment reveals that both diversity and density exhibit clear spatial patterns of soil seed banks but differ in their environmental determinants. These findings alone do not necessarily mean that this biodiversity reservoir has strong buffering capacity under climate change, because both climate and soil conditions influence seed bank diversity and density. Based on a large number and long history of studies globally, we provide quantitative evidence of how environmental conditions shape soil seed bank distributions and spatially explicit maps of this biodiversity reservoir in plant communities worldwide. Our quantification of environmental determinants and global mapping can be readily applied to dynamic global vegetation and plant diversity models to enable a more complete and accurate prediction of the impact of ongoing environmental changes on plant diversity (both above- and belowground) at the global scale. The next research challenge will be to plot current (visible) aboveground plant diversity (ideally using the available data in the studies themselves) against soil seed bank diversity under global change scenarios in order to pinpoint even more accurately which plant communities, ecosystems and biomes (and their turn-over) are most at risk of losing their diversity due to global changes.

## Methods

**Global data of soil seed bank**. To identify published studies on soil seed banks worldwide, we conducted an ISI Web of Science search covering the time period

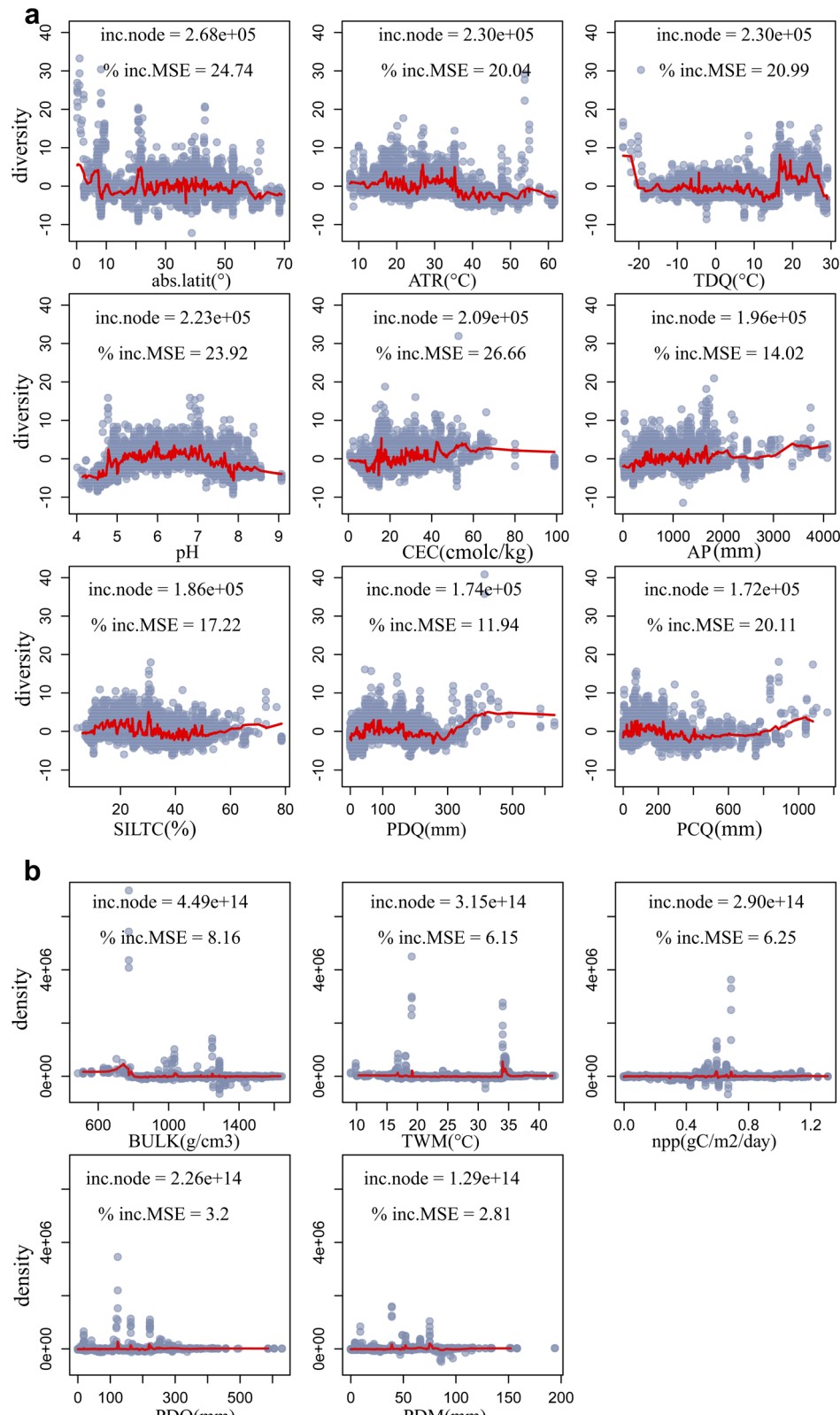

**Fig. 3 Partial feature contributions (the marginal effect of a variable on response) of the most important variables for soil seed banks. a** diversity; **b** density. Variable importance (inc. node) is the decrease in the residual sum of squares that results from splitting regression trees using the variable. The percentage increase in mean squared error (% inc. MSE) is the increase in model error as a result of randomly shuffling the order of values in the vector. abs.latit, absolute latitude; AP, annual precipitation; ATR, annual temperature range; BULK, bulk density; CEC, cation exchange capacity; npp, plant productivity (net primary production); PCQ, precipitation of coldest quarter; PDM, precipitation of driest month; PDQ, precipitation of driest quarter; pH, pH measured in water; SILTC, silt (mass %); TDQ, mean temperature of driest quarter; TWM, max temperature of warmest month.

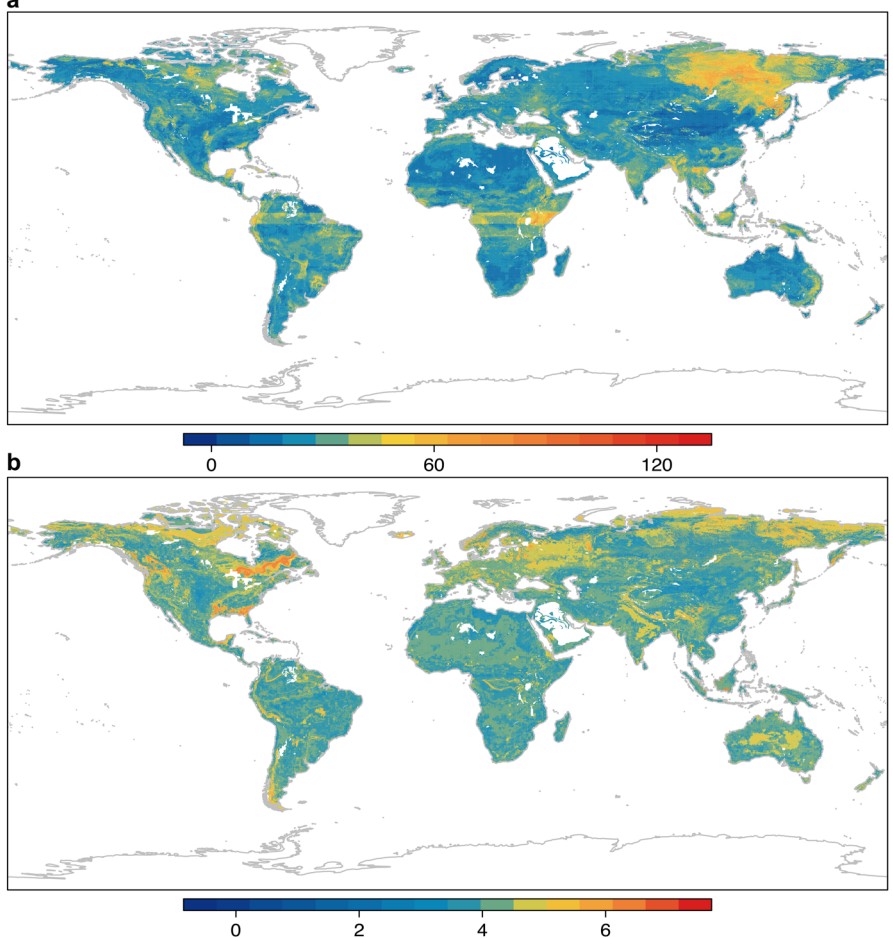

**Fig. 4 Extrapolated global maps of soil seed banks. a** diversity in terms of number of species per $0.01\,\text{m}^2$; **b** density as number of seeds per $\text{m}^2$. In **b**, values are log10-transformed to facilitate viewing. The spatial resolution of grid cells is 5 arcmin-by-5 arcmin.

from 1900 onwards using the following search terms: ("soil seedbank" OR "soil seed bank" OR "soil propagule bank" OR "soil stored seed" OR "buried viable seed") AND (composition OR richness OR diversity OR "species number" OR density OR abundance). We updated the search several times during the last few years, and the latest update was in May 2021. The total return was 2,166 publications. In addition, we conducted a literature search in the China National Knowledge Infrastructure (CNKI) to identify publications in Chinese. The abstract of each publication was read individually to assess suitability of the study before obtaining the publication, and the reference list of each publication collected was inspected to identify additional relevant publications. Finally, we pre-selected a total of 1774 publications on soil seed banks worldwide (1,472 in English and 302 in Chinese).

To avoid bias in publication selection, only those studies were selected that met all of the following criteria. (1) Samples were collected from natural vegetation, and the results reported at least one data point on diversity and/or density of the soil seed bank. Old-fields abandoned for longer than five years were considered because they resemble natural vegetation, while weed/crop experiments were not included because agricultural seed banks reflect cultivation and cropping patterns and thus any environmental control is secondary. (2) Studies were included only when diversity or density were measured at the whole community level (i.e., all species in a community). (3) Only studies conducted in terrestrial ecosystems were included. In total, 1,502 publications met the above criteria (Supplementary Data 1 and 2).

For studies that included different levels of natural gradients (e.g., different ecosystems, soil depth, sampling time or topographic and moisture gradients), data for these levels were considered as independent. If environmental conditions were manipulated in a study (e.g., herbivory, nutrients, warming or $CO_2$), we extracted only data from the treatment that most closely reflected the situation under natural conditions. In addition, we excluded review/synthetic papers and used only studies that reported primary field data. We extracted data from the text, tables, digitized graphs and supplementary materials.

**Statistical analyses**. All statistical analyses were performed with the open-source language R (version 3.4.3, https://cran.r-project.org/).

Sampling time, area and soil depth differed both within and among studies, which might induce biases when comparing data. To account for such biases, we did three things. First, we divided the data into different seasons according to sampling time (i.e., spring, summer, autumn and winter). Then, we standardized all data to the season with the highest value (winter) by calculating an average ratio between that season and winter and then multiplying by that ratio. In this way, we standardized all data to the time with highest value, which made all data comparable across different sampling times. Because tropical regions have low seasonality, data collected from (sub)tropical biomes were not standardized using this procedure. Second, we used a species-area curve (Eq. (1)) to account for the difference between sampling area for diversity[6]:

$$S = \text{C}A^Z, \qquad (1)$$

where $S$ is the number of species (diversity), $C$ a fitted constant, $A$ the sampling area and $Z$ a fitted constant. We used pooled diversity data to estimate the parameters and standardized all diversity to the most commonly reported area ($0.01\,\text{m}^2$). To minimize the bias caused by extreme values, the outliers in the data were identified by Rosner's test using EnvStats package[30], and outliers above the upper limit were capped with the value of the 95th percentile. Notably, the species–area relationship could have considerable geographical variation due to biomes[31]; thus, we modeled the species-area curve for each biome separately. For this, we extracted the biome type of each data point from the Terrestrial Ecoregions of the World (TEOW)[32]. Most studies reported density as number of seeds per $\text{m}^2$ of soil surface. Otherwise, we used sample area to extrapolate data to number of seeds per $\text{m}^2$. Third, we used linear regression models to determine relationships between seed bank data and upper and lower boundaries of sampling soil depths (slices), and estimated parameters were used to standardized data to the most commonly reported soil depth (0–5 cm), which made the data comparable. To account for the differences among biomes, we modeled these relationships for each biome separately. Further, to determine whether there was potential artifact of sampling bias, we compared seed bank diversity and density for each biome between Southern and Northern Hemisphere (Supplementary Table 5). Of the 9 comparisons for diversity, only 4 pairs are significantly different, among which 3 pairs actually have higher value in the Southern Hemisphere. For density, mean

values were also not biased towards the Southern or Northern Hemisphere. These results clearly indicate that our global predictions of the higher soil seed banks in the Northern Hemisphere (Fig. 4) are unlikely to reflect an artifact of sampling bias between the Northern and Southern Hemispheres. We then used non-parametric Kruskal–Wallis tests to compare the differences in soil seed banks among biome types.

Spatial autocorrelation in primary data can lead to overoptimistic assessment of model predictive power[33,34]. To account for this issue, we computed semivariograms to determine spatial autocorrelation patterns in our data prior to spatial analyses. To identify the key factors that determine the pattern of soil seed banks worldwide, we selected 31 global predictors previously reported to affect plant performance[16–18]: 19 climatic indices, 8 top soil variables, 1 human footprint (a composite variable compiled on eight variables measuring the direct and indirect human pressures on the environment globally[35]), 2 plant indices and 1 spatial coordinate (see Table S1 for sources of the predictors).

We implemented the random-forest algorithm to model the relationships between these predictors and soil seed banks. The random-forest model is a data-driven ensemble learning approach that averages over multiple regression trees, each of which uses a random subset of all the model variables to predict a response[36]. Random-forest handles highly collinear predictors by spreading the importance of the variable across all variables[37]. This approach runs efficiently on large data bases and has been successfully applied to global analyses[17,18]. We first determined the influence of all 31 predictors on soil seed banks. Variable importance was ranked in terms of the increase in node purity, which is the decrease in the residual sum of squares that results from splitting regression trees using the variable. We also reported the percentage increase in mean squared error (MSE), which quantifies the increase in model error as a result of randomly shuffling the order of values in the vector. The random-forest algorithm was carried out using the R package randomForest[38]. The full models (using all 31 predictors) were run using 100 regression trees each.

We then implemented a variable selection procedure using the R package VSURF[39], which used the random forests permutation-based score of importance and proceeded using a stepwise forward strategy for variable introduction. Specifically, a variable was added only if the decrease in error was larger than a threshold, i.e., the decrease in out-of-bag (OOB) error had to be significantly greater than the average variation obtained by adding noisy variables. The most important predictor variables for seed bank diversity and density were selected to build final models (Supplementary Fig. 7). We ran the random-forest algorithm using the final models. We found that final models explained higher variance than full models (Supplementary Table 4). We plotted the final variable response of soil seed bank to each of the most important predictors using the R package forestFloor[40].

To test the sensitivity of final model performance, we performed K-fold cross-validations that test the sensitivity of model predictions to the exclusion of random subsets from the training data. Cross-validation was implemented using the R package rfUtilities[41]. We ran 99 iterations that withheld 10% of the model training data. These tests showed that our training data had sufficient redundancy to ensure that our model conclusions were robust.

Finally, we derived global predictions of diversity and density of soil seed banks in the spatial resolution of grid cell of 5 arcmin-by-5 arcmin. We made predictions based on the final random-forest models and by using the same predictor variables for the global grid.

**Reporting summary**. Further information on research design is available in the Nature Research Reporting Summary linked to this article.

## Data availability
Data and references from which data were collected supporting the findings of this study are available in the Supplementary Data 1 and 2. Terrestrial Ecoregions of the World (TEOW) are publicly available on the World Wildlife Fund (WWF) website [https://www.worldwildlife.org/]. Climate data reported in this study are publicly available on the WorldClim database [https://www.worldclim.org/]. Soil data are publicly available on the SoilGrids system [https://soilgrids.org/].

## Code availability
The R codes used for analyses are available in the Supplementary Software.

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

## Acknowledgements

We thank Prof. Ken Thompson from the University of Sheffield, UK, for critical comments, which improved the manuscript. We thank W. Zhang, Y. Xu, C. Di, W. Ji, M. Dong, W. Ren, Y. Yang, T. Shao, and M. Wu from the School of Life Sciences, Shanxi Normal University for the assistance in collecting part of the data. This work was supported by the National Natural Science Foundation of China (32071524 and 31770514 to X.Y., and 31861143024 to Z.H.). International research travel by J.H.C.C. was partly funded by the Royal Netherlands Academy of Arts and Sciences (KNAW, CEP grant 12CDP007).

## Author contributions

X.Y., C.C.B., J.M.B., Z.H., and J.H.C.C. conceived the study. X.Y. and R.G. collected the data. X.Y. performed the analyses. The manuscript was drafted by X.Y., with contributions from C.C.B., J.M.B., R.J.P., and J.H.C.C.

## Competing interests

The authors declare no competing interests.
