## [Peer Review File · Nature Communications]

Global patterns of potential future plant diversity hidden in soil seed banksPeer Review File

Reviewer #1 (Remarks to the Author):

This study intends to fill important knowledge gaps regarding the global patterns in the density and diversity of soil seed banks. Such global analyses don't exist, so the work is certainly pioneering. The Authors gathered a comprehensive dataset from a comprehensive literature search. I highly appreciate this effort. I have checked the list of references and I found it very comprehensive. The Authors aim to derive global trends from the completed database. This is a great effort, but I have some concerns regarding the global extrapolations, the grain sizes and the appropriateness of the species richness standardization (please see below). Therefore, I think that some of the conclusions need to be more careful and/or supported by further analyses. Please see below my comments.

L. 165-177: Fig. 4 shows nice spatial trends that can be derived from the existing data; however I am not sure whether such generalizations are possible at the global scale (e.g. a certain part of a certain continent is characterised by higher or lower seed densities than other parts of other continents). Such patterns are modified by several parameters that cannot be captured at the global scale. Most importantly, in many seed bank studies, not the zonal vegetation is studied – e.g. small meadows (1%) are studied in a matrix of boreal forest (99%) -> generalizing the findings of such studies to a broader region is biased. In many regions, research interest in seed bank studies is focussed not on the dominant vegetation type, but on rare habitats (because researchers are interested whether seed bank can contribute to the restoration of the rare habitats). I suggest to be more cautious about the global conclusions of seed bank density and diversity when interpreting the results from the map.

L. 177-181: I suggest being a bit more careful with this prognosis or mentioning that these are based purely on total density and diversity records. A high density or high diversity seed bank does not necessarily guarantee community resilience, as soil seed bank often consists of non-native species, or disturbance-tolerant or weedy species. Therefore, having a rough estimation on the density and diversity of soil seed bank in different ecosystems is a very important first step, but not sufficient for drawing conclusions for the restoration potential of seed banks.

You mention later that the next step would be to compare the similarity of the vegetation and seed bank – I agree, that would be great. However I think it would be only possible by using the available data in the studies themselves (please see my comments on grain size) and not by using global vegetation data.

L. 284: The first criterion, i.e. the samples have to originate from natural vegetation is in contradiction with the fact that you also included studies on old-fields. Please define what was considered as natural vegetation. Also, criteria 1 and 4 might be merged (?)

Please also give a reference for the characterization of the studied ecosystem types. These are quite broad categories, e.g. forest can vary from tropical rainforest to taiga; grassland can vary from open dry grassland to tall meadows...

L. 304-306: This standardization for spring-summer-autumn-winter works for many parts of the world, but probably not for regions with fewer seasons, or rainy and dry seasons. How did you treat such cases?

L. 309-317: I am not completely convinced about the standardization of the species richness data. For the aboveground vegetation, a large body of literature is available on species-area relationships, even though these relationships can be very different across biomes. For soil seed bank, there is very few information on the spatial patterns of the species; in some ecosystems, seed bank shows an aggregated pattern, in others it is distributed more homogeneously. I am not completely convinced whether species richness of soil seed bank can be standardized to a certain grain size without considerable bias.

Fig. 4: The extrapolated map looks nice, but it seems to be a little bit too brave to extrapolate seed bank density and diversity e.g. for the whole territory of Russia, when there are only

approximately 10 data points available for the whole country according to Fig. 1. As many seed bank studies were conducted in azonal or extrazonal vegetation, there can be considerable bias in the extrapolation (please see my other comment on Extended Data table 1).

In general, the quality of the supplementary figures could be improved a little bit. It is not a crucial problem, but a nicer layout would fit more in such a prestigious journal.

Extended Fig. 2: Please add the units of measurement to the horizontal axis. Is it a log-transformed axis?

Extended Fig. 4: Please add the units of measurements to the horizontal axes on both panels. Also, please add density and diversity to the vertical axes, it helps the reader a lot.

Extended Data table 1: I wonder whether it is possible to derive the soil variables properly at the resolution of the study sites. In many studies, azonal or extrazonal vegetation is studied, which is often formed on soils different from the dominant soil type in a region.

Reviewer #2 (Remarks to the Author):

This study gathered a large dataset of seed bank data in various biomes with a global distribution. It provides a statistical analysis on this large dataset to select the best environmental predictors of empirical variations in seed bank density and diversity. The work performed to gather this dataset is substantial, but the methodology of the study has some major limitations that need to be addressed. Also, the biological insights drawn from the performed analysis have not been fully thought through.

1-Separate analysis of diversity and density of seed banks

The study separately analyzes seed bank density and diversity while there is a clear dependence of diversity on density. Alternative statistical analyses may be more appropriate to better tease apart direct and indirect drivers of seed bank diversity. Also, from a more technical perspective, complementary metrics of diversity may be explored rather than the single species richness.

2-Selection of predictors

The manuscript does not provide the reader with any biological rationale guiding the choice of environmental predictors. The authors seem to have tested an arbitrary set of easily available environmental variables without prior thinking (this may sound a little harsh, but this is the impression the reader gets from the reading). Second, there is no a priori hypotheses grounded on biology on the likely effect of each environmental predictor tested on seed bank density and diversity. An alternative presentation starting from a carefully justified set of hypotheses that would be tested with the dataset would be much more interesting and informative. Third, there is no biotic predictors such as regional pool diversity, disturbance regime/history, biome identity or site productivity, that may constitute important drivers of seed bank characteristics. Including such complementary predictors would considerably strengthen the manuscript. (and there are some global data on such variables or proxies of these variables).

3-Rationale for using random-forest models

The manuscript should justify the use of random-forest models. Why not choosing simpler models like linear or polynomial ones to assess qualitative relationships that could be biologically interpreted? Looking at the fits in Figure 3, it seems that the random forest model is more interpolating some noise in the data than revealing any biologically meaningful trend.

4-Questionable standardization protocols

The authors rightly devised standardization procedures to make data comparable across studies. Unfortunately, the standardization procedures seem awkward and might be improved.

-seasonality: the seasonality patterns in seed bank density is likely to be variable across regions (for instance temperate vs tropical areas) or biomes (e.g., forests vs tundra). The seasonality correction factor should thus be more specific to each situation rather than computed by pooling all the dataset.

-sampling area: again, using a single relationship for the whole dataset is clumsy, since the species-area curve is likely to depend on seed bank density: a denser seed bank is likely to more

quickly saturate with area than a less dense bank. Also, the treatment of outliers is not appropriately justified with references from the statistical literature (the procedure looks weird to me, but I'm not an expert on this topic).

-sampling depth: the description of the procedure should be clarified. Again, pooling the whole dataset looks weird, since the relationship between sampling depth and seed bank density is likely to vary between biomes and according to local soil depth and/or disturbance regime.

All in all, additional analyses should be performed to choose the right standardization method for each variable.

5-Additional methodological questions/suggestions

-Were the data log-transformed before analysis (for seed bank density)? This needs to be stated. If yes, I am surprised that there are still outliers. Some discussion on these outliers may be added to clarify this.

-the methods used for the determination of seed banks need to be informed in the dataset (seed counts vs germination tests...)

-for the variable selection procedure, which criteria was used to stop the forward addition of variables? How were variables chosen at each step of the stepwise forward procedure?

-R-squared should be reported (or any other measure of goodness of fit).

-R code and the dataset should be included as supplementary information for reproducibility reasons.

Minor comments:

-The study reports a spatial bias in the data with stronger sampling of some continents: are there potential consequences of this bias? This needs to be discussed.

-The study does not report the variance explained by the prediction model, while this is a key element to appraise the results.

-The discussion on the global scale patterns lacks some consideration of the possible effect of biome turn-over at these scales.

-Lines 177-178. The discussion on this topic may be enriched by considering the temporal integration of seed bank effects for long-lived trees, and the possible biased vision of seed banks in forests due to masting events.

-Lines 183-186. This sentence is unclear.

-Lines 189-193: how exactly these results could be mobilized?

-Lines 193-196: the rationale of what is proposed is unclear.

-Figure 2: define the meaning of the abbreviations in the main text or in the figure legend.

-Figure 3: what is plotted in the Y-axis?

Reviewer #3 (Remarks to the Author):

This is a simple yet elegant study that quantifies global patterns and determinants of soil seed banks. The authors make use of the wider literature to collate an impressive dataset that covers a wide latitudinal breadth, although there appears to be the usual North-temperate bias in data availability.

The methodologies are what I would expect from a basic macroecological analysis, such as ensuring congruency among different datasets, consideration of spatial autocorrelation and variation in unit area. I liked that the authors also searched for studies outside of the English language to include in their dataset.

For the analysis, the authors opted to build random forest models, which makes sense given the large number of predictor variables. Their findings show variation in soil seed banks among ecosystem types, with variables such as absolute latitude being of particular importance.

Overall, I thoroughly enjoyed reading this manuscript and I believe it will be of broad interest to the scientific community. I do, however, have a couple of concerns that should be addressed:

1. Standardisation of region area. SARs are not uniform among biomes, and thus the authors should account for this geographical variation. See Gerstner et al 2013 "Accounting for geographical variation in species-area relationships improves the prediction of plant species richness at the global scale".

2. Biodiversity gaps. I understand that in such a global assessment there will always be data gaps.

However, they must be discussed at the very least, or there should be some kind of assessment for data quality, or even better attempts to fill these gaps. For example, could it be that the result that the higher density of seed banks in the Northern Hemisphere is simply an artefact of sampling bias? How can we believe this result when we look at the map of data locations (Fig 1.) and see these large empty spaces in the tropics and Global South?

3. I could have read over it but to me it is not fully clear how you go from Fig 1. to Fig 4. If I look at Russia, which has hardly any soil seed bank data available according to Fig 1, yet then has quite a high density of seeds according to Fig 4, I get confused. Please make this clearer in text.

Thanks again for the interesting study and I hope the comments help.

Warm regards,
Dr. Amanda Taylor

We sincerely appreciate the insightful comments provided by the three reviewers. We believe that our manuscript has become clearer and more robust as a result of the review process. Please find the original comments in black, followed by our responses in blue.

REVIEWER COMMENTS

Reviewer #1 (Remarks to the Author):

This study intends to fill important knowledge gaps regarding the global patterns in the density and diversity of soil seed banks. Such global analyses don't exist, so the work is certainly pioneering. The Authors gathered a comprehensive dataset from a comprehensive literature search. I highly appreciate this effort. I have checked the list of references and I found it very comprehensive. The Authors aim to derive global trends from the completed database. This is a great effort, but I have some concerns regarding the global extrapolations, the grain sizes and the appropriateness of the species richness standardization (please see below). Therefore, I think that some of the conclusions need to be more careful and/or supported by further analyses.

Response: Thank you for your careful and favorable reading of our paper. We agree with your comments on global extrapolations, grain sizes and the appropriateness of the species richness standardization and thus have conducted additional analyses and revised our conclusion accordingly (please see below).

Please see below my comments.

L. 165-177: Fig. 4 shows nice spatial trends that can be derived from the existing data; however I am not sure whether such generalizations are possible at the global scale (e.g. a certain part of a certain continent is characterised by higher or lower seed densities than other parts of other continents). Such patterns are modified by several parameters that cannot be captured at the global scale. Most importantly, in many seed bank studies, not the zonal vegetation is studied – e.g. small meadows (1%) are studied in a matrix of boreal forest (99%) -> generalizing the findings of such studies to a broader region is biased. In many regions, research interest in seed bank studies is focussed not on the dominant vegetation type, but on rare habitats (because researchers are interested whether seed bank can contribute to the restoration of the rare habitats). I suggest to be more cautious about the global conclusions of seed bank density and diversity when interpreting the results from the map.

Response: Thank you for the appreciation for Fig. 4. We agree that studies in azonal vegetation or in rare habitats could bias the generalization at the global scale. During data collection, we tried our best to collect data representing the dominant vegetation type of that region; thus, our database mostly represents the soil seed banks of the dominant vegetation type(s). Yet, we are aware that our global prediction is not

completely free of the influences of azonal vegetation and rare habitats. To address your concerns, we have added the following statement to the text (lines 212-216):

However, our global assessment should be interpreted with caution since some studies in azonal vegetation or in rare habitats in our database did not fully reflect soil seed banks in that region, and thus these data shortcomings may have induced bias in our global predictions.

L. 177-181: I suggest being a bit more careful with this prognosis or mentioning that these are based purely on total density and diversity records. A high density or high diversity seed bank does not necessarily guarantee community resilience, as soil seed bank often consists of non-native species, or disturbance-tolerant or weedy species. Therefore, having a rough estimation on the density and diversity of soil seed bank in different ecosystems is a very important first step, but not sufficient for drawing conclusions for the restoration potential of seed banks.

Response: We agree and have revised this section on lines 220-223:

Nevertheless, based on our global patterns of soil seed bank diversity and density, the latitudinal pattern strongly suggests that the biodiversity of (sub-)tropical forests is particularly vulnerable to large-scale climatic or land-use disturbances.

You mention later that the next step would be to compare the similarity of the vegetation and seed bank – I agree, that would be great. However I think it would be only possible by using the available data in the studies themselves (please see my comments on grain size) and not by using global vegetation data.

Response: We agree and have revised this sentence on lines 245-250:

The next research challenge will be to plot current (visible) aboveground plant diversity (ideally using the available data in the studies themselves) against soil seed bank diversity under global change scenarios in order to pinpoint even more accurately which plant communities, ecosystems and biomes (and their turn-over) are most at risk of losing their diversity due to global changes.

L. 284: The first criterion, i.e. the samples have to originate from natural vegetation is in contradiction with the fact that you also included studies on old-fields. Please define what was considered as natural vegetation. Also, criteria 1 and 4 might be merged (?)

Response: According to your comments, we have added more information about old-fields and have merged criteria 1 and 4 (lines 338-343):

(1) Samples were collected from natural vegetation, and the results reported at least one data point on diversity and/or density of the soil seed bank. Old-fields abandoned for longer than five years were considered because they resemble natural vegetation, while weed/crop experiments

were not included because agricultural seed banks reflect cultivation and cropping patterns and thus any environmental control is secondary.

Please also give a reference for the characterization of the studied ecosystem types. These are quite broad categories, e.g. forest can vary from tropical rainforest to taiga; grassland can vary from open dry grassland to tall meadows...

Response: We have changed to the 14 biomes defined by Olson et al. (Supplementary Fig. 4).

Reference

32. Olson D. M., et al. Terrestrial ecoregions of the world: A new map of life on Earth. *BioScience* 51, 933–938 (2001).

L. 304-306: This standardization for spring-summer-autumn-winter works for many parts of the world, but probably not for regions with fewer seasons, or rainy and dry seasons. How did you treat such cases?

Response: We have divided the standardization for season based on biome, and the relevant information has been provided in lines 367-368:

Because tropical regions have low seasonality, data collected from (sub)tropical biomes were not standardized using this procedure.

L. 309-317: I am not completely convinced about the standardization of the species richness data. For the aboveground vegetation, a large body of literature is available on species-area relationships, even though these relationships can be very different across biomes. For soil seed bank, there is very few information on the spatial patterns of the species; in some ecosystems, seed bank shows an aggregated pattern, in others it is distributed more homogeneously. I am not completely convinced whether species richness of soil seed bank can be standardized to a certain grain size without considerable bias.

Response: We have improved the standardization of the species richness data. Relevant information has been provided on lines 377-380:

Notably, the species–area relationship could have considerable geographical variation due to biomes³¹; thus, we modeled the species-area curve for each biome separately. For this, we extracted the biome type of each data point from the Terrestrial Ecoregions of the World (TEOW)³².

Fig. 4: The extrapolated map looks nice, but it seems to be a little bit too brave to extrapolate seed bank density and diversity e.g. for the whole territory of Russia, when there are only approximately 10 data points available for the whole country according to Fig. 1. As many seed bank studies were conducted in azonal or extrazonal vegetation, there can be considerable bias in the extrapolation (please see my other comment on Extended Data table 1).

Response: We assessed seed bank diversity and density at the global scale, based on

the available data which reflected how many studies have been conducted in different countries. In addition, the data presented in Fig. 1 were used for finding relationships between soil seed bank and environmental variables, from which the extrapolated map was drawn. Therefore, the extrapolated map did not rely only on the data points for a particular country. We agree that studies in azonal or extrazonal vegetation could induce bias in our global prediction, and this has now been discussed (please see above).

In general, the quality of the supplementary figures could be improved a little bit. It is not a crucial problem, but a nicer layout would fit more in such a prestigious journal.

Response: We have redrawn all supplementary figures.

Extended Fig. 2: Please add the units of measurement to the horizontal axis. Is it a log-transformed axis?

Response: The units of the horizontal axis in Extended Fig. 2 have been added.

Extended Fig. 4: Please add the units of measurements to the horizontal axes on both panels. Also, please add density and diversity to the vertical axes, it helps the reader a lot.

Response: The units have been added to the horizontal axis and density and diversity to the vertical axes in Extended Fig. 4 (now Supplementary Figure 3).

Extended Data table 1: I wonder whether it is possible to derive the soil variables properly at the resolution of the study sites. In many studies, azonal or extrazonal vegetation is studied, which is often formed on soils different from the dominant soil type in a region.

Response: We agree that azonal or extrazonal vegetation could induce bias in our global predictions (please see above). For soil variables, we used the most recent global soil database (soil grid) to model the relationships between soil environment and seed bank. As shown in Fig. 2, the strongest predictor of soil seed bank was absolute latitude; thus, we think that the effect of soil variables, if it is impossible to derive them properly from azonal vegetation, should be secondary.

Reviewer #2 (Remarks to the Author):

This study gathered a large dataset of seed bank data in various biomes with a global distribution. It provides a statistical analysis on this large dataset to select the best environmental predictors of empirical variations in seed bank density and diversity. The work performed to gather this dataset is substantial, but the methodology of the study has some major limitations that need to be addressed. Also, the biological insights drawn from the performed analysis have not been fully thought through.

Response: Thank you for your appreciation of our work. We have carefully addressed the major limitation in the methodology and performed additional analyses according

to your comments (please see below).

1-Separate analysis of diversity and density of seed banks

The study separately analyzes seed bank density and diversity while there is a clear dependence of diversity on density. Alternative statistical analyses may be more appropriate to better tease apart direct and indirect drivers of seed bank diversity. Also, from a more technical perspective, complementary metrics of diversity may be explored rather than the single species richness.

Response: We agree that soil seed bank diversity could be related to density. Yet, as the first step towards a global assessment of soil seed bank, the aim of our work was to determine how the two aspects of soil seed bank (diversity and density) were related to environmental conditions. Therefore, we chose to analyze diversity and density separately. However, we agree that the dependence of diversity on density certainly is an interesting topic worthy of further study, and this point has been added in lines 229-232:

Further, our analyses suggest that the least vulnerable ecosystems in terms of hidden diversity should be those that combine high seedbank diversity with high density; and therefore the relationships between the two variables across the global map certainly would be an interesting topic worthy of further study.

For diversity, we used species richness, which is the most commonly used index for diversity and the most intuitive one that can be easily understood by readers. We agree that complementary metrics of diversity (e.g. functional diversity, Shannon diversity) also are important measures for diversity.

2-Selection of predictors

The manuscript does not provide the reader with any biological rationale guiding the choice of environmental predictors. The authors seem to have tested an arbitrary set of easily available environmental variables without prior thinking (this may sound a little harsh, but this is the impression the reader gets from the reading). Second, there is no a priori hypotheses grounded on biology on the likely effect of each environmental predictor tested on seed bank density and diversity. An alternative presentation starting from a carefully justified set of hypotheses that would be tested with the dataset would be much more interesting and informative. Third, there is no biotic predictors such as regional pool diversity, disturbance regime/history, biome identity or site productivity, that may constitute important drivers of seed bank characteristics. Including such complementary predictors would considerably strengthen the manuscript. (and there are some global data on such variables or proxies of these variables).

Response: To address your concerns, we have done three things. First, we have provided the rationale guiding the choice of environmental predictors. Second, following your comment we have added clearer hypotheses on lines 59-67:

Close relationships between the soil seed banks and environmental variables (including climate and soil) have been reported, albeit mostly at the local or regional scale^{5,9,10}. We hypothesized that soil seed bank composition and density should show clear global patterns since environmental conditions vary geographically across the Earth. Biologically, since seed dormancy and longevity in the soil are determined by temperature, precipitation and soil environments¹ and climate and soil drive plant productivity that in turn should drive seed influx into the soil, we further hypothesized that climate and soil variables are important for predicting soil seed bank diversity and density at the global scale.

Third, we have added regional pool diversity (global map of vascular plants) and site productivity (global biomass) to our predictors, and results and figures have been reworked accordingly.

3-Rationale for using random-forest models

The manuscript should justify the use of random-forest models. Why not choosing simpler models like linear or polynomial ones to assess qualitative relationships that could be biologically interpreted? Looking at the fits in Figure 3, it seems that the random forest model is more interpolating some noise in the data than revealing any biologically meaningful trend.

Response: The random-forest model is a data-driven ensemble learning approach that averages over multiple regression trees, each of which uses a random subset of all the model variables to predict a response. Random-forest handles highly collinear predictors by spreading the importance of the variable across all variables and is commonly used for analyzing a large database at the global scale. Therefore, the random-forest model is more suitable for analyzing our database than linear or polynomial ones. In fact, Figure 3 shows the partial feature contribution of each selected variable, not the fit of the full model.

4-Questionable standardization protocols

The authors rightly devised standardization procedures to make data comparable across studies. Unfortunately, the standardization procedures seem awkward and might be improved.

-seasonality: the seasonality patterns in seed bank density is likely to be variable across regions (for instance temperate vs tropical areas) or biomes (e.g., forests vs tundra). The seasonality correction factor should thus be more specific to each situation rather than computed by pooling all the dataset.

-sampling area: again, using a single relationship for the whole dataset is clumsy, since the species-area curve is likely to depend on seed bank density: a denser seed bank is likely to more quickly saturate with area than a less dense bank. Also, the treatment of outliers is not appropriately justified with references from the statistical literature (the procedure looks weird to me, but I'm not an expert on this topic).

-sampling depth: the description of the procedure should be clarified. Again, pooling the whole dataset looks weird, since the relationship between sampling depth and seed bank density is likely to vary between biomes and according to local soil depth and/or disturbance regime.

All in all, additional analyses should be performed to choose the right standardization method for each variable.

Response: According to your suggestions, we have substantially revised all standardization protocols (lines 362-387).

First, we divided the data into different seasons according to sampling time (i.e. spring, summer, autumn and winter). Then, we standardized all data to the season with the highest value (winter) by calculating an average ratio between that season and winter and then multiplying by that ratio. In this way, we standardized all data to the time with highest value, which made all data comparable across different sampling times. Because tropical regions have low seasonality, data collected from (sub)tropical biomes were not standardized using this procedure. Second, we used a species-area curve (eqn. 1) to account for the difference between sampling area for diversity⁶:

$$S = CA^Z \quad , \quad \text{eqn. 1}$$

where S is the number of species (diversity), C a fitted constant, A the sampling area and Z a fitted constant. We used pooled diversity data to estimate the parameters and standardized all diversity to the most commonly reported area (0.01 m²). To minimize the bias caused by extreme values, the outliers in the data were identified by Rosner's test using EnvStats package³⁰, and outliers above the upper limit were capped with the value of the 95th percentile. Notably, the species–area relationship could have considerable geographical variation due to biomes³¹; thus, we modeled the species-area curve for each biome separately. For this, we extracted the biome type of each data point from the Terrestrial Ecoregions of the World (TEOW)³². Most studies reported density as number of seeds per m² of soil surface. Otherwise, we used sample area to extrapolate data to number of seeds per m². Third, we used linear regression models to determine relationships between seed bank data and upper and lower boundaries of sampling soil depths (slices), and estimated parameters were used to standardized data to the most commonly reported soil depth (0-5 cm), which made the data comparable. To account for the differences among biomes, we modeled these relationships for each biome separately.

-Were the data log-transformed before analysis (for seed bank density)? This needs to be stated. If yes, I am surprised that there are still outliers. Some discussion on these outliers may be added to clarify this.

-the methods used for the determination of seed banks need to be informed in the dataset (seed counts vs germination tests...)

-for the variable selection procedure, which criteria was used to stop the forward addition of variables? How were variable chosen at each step of the stepwise forward procedure?

-R-squared should be reported (or any other measure of goodness of fit).

-R code and the dataset should be included as supplementary information for reproducibility reasons.

Response: -Seed bank density was not log-transformed before analysis.

-We have now provided the methods used for the determination of seed banks in the dataset.

-We have added more information for the variable selection procedure (lines 421-424):

Specifically, a variable was added only if the decrease in error was larger than a threshold, i.e. the decrease in out-of-bag (OOB) error had to be significantly greater than the average variation obtained by adding noisy variables.

- R-squared has been reported in Supplementary Table 4.

- R code and the dataset have now been provided in Supplementary Data.

Minor comments:

-The study reports a spatial bias in the data with stronger sampling of some continents: are there potential consequences of this bias? This needs to be discussed.

Response: We have added the following to the discussion (lines 216-220):

Moreover, data gaps in our database are also likely to have had an effect on the global predictions, i.e. fewer data available from some continents (e.g. northern Asia and Africa) could lead to less confidence for prediction in these regions. For example, Russia has very few soil seed bank data, which may have led to an inaccurate prediction for this country.

-The study does not report the variance explained by the prediction model, while this is a key element to appraise the results.

Response: We have reported the variance explained by the prediction model in Supplementary Table 4.

-The discussion on the global scale patterns lacks some consideration of the possible effect of biome turn-over at these scales.

Response: We have added biome turn-over on line 249.

-Lines 177-178. The discussion on this topic may be enriched by considering the temporal integration of seed bank effects for long-lived trees, and the possible biased vision of seed banks in forests due to masting events.

Response: We have added such effects (lines 220-226):

Nevertheless, based on our global patterns of soil seed bank diversity and density, the latitudinal pattern strongly suggests that the biodiversity of (sub-)tropical forests is particularly vulnerable to large-scale climatic or land-use disturbances. However, in-depth investigation is needed to quantify the extent to which temporal integration of seed bank effects for long-lived trees and seed masting events may buffer the effects of low seed bank diversity and density at any given time of sampling.

-Lines 183-186. This sentence is unclear.

Response: We have clarified this sentence as (lines 234-238):

These findings alone do not necessarily mean that this biodiversity reservoir has strong buffering capacity under climate change, because both climate and soil conditions influence seed bank diversity and density, and global environmental change will therefore have deleterious impact on soil seed banks.

-Lines 189-193: how exactly these results could be mobilized?

Response: We have clarified this sentence as (lines 241-245):

Our quantification of environmental determinants and global mapping can be readily applied to dynamic global vegetation and plant diversity models to enable a more complete and accurate prediction of the impact of ongoing environmental changes on plant diversity (both above- and belowground) at the global scale.

-Lines 193-196: the rationale of what is proposed is unclear.

Response: We have added the rationale and clarified this sentence as (lines 245-250):

The next research challenge will be to plot current (visible) aboveground plant diversity (ideally using the available data in the studies themselves) against soil seed bank diversity under global change scenarios in order to pinpoint even more accurately which plant communities, ecosystems and biomes (and their turn-over) are most at risk of losing their diversity due to global changes.

-Figure 2: define the meaning of the abbreviations in the main text or in the figure legend.

Response: We have defined the meaning of the abbreviations in the main text.

-Figure 3: what is plotted in the Y-axis?

Response: Y-axis is diversity in Fig. 3a and density in Fig. 3b, and now they have been added to the figure.

Reviewer #3 (Remarks to the Author):

This is a simple yet elegant study that quantifies global patterns and determinants of soil seed banks. The authors make use of the wider literature to collate an impressive dataset that covers a wide latitudinal breadth, although there appears to be the usual North-temperate bias in data availability.

The methodologies are what I would expect from a basic macroecological analysis, such as ensuring congruency among different datasets, consideration of spatial autocorrelation and variation in unit area. I liked that the authors also searched for studies outside of the English language to include in their dataset.

For the analysis, the authors opted to build random forest models, which makes sense given the large number of predictor variables. Their findings show variation in soil seed banks among ecosystem types, with variables such as absolute latitude being of particular importance.

Response: Thank you for your appreciation of our work.

Overall, I thoroughly enjoyed reading this manuscript and I believe it will be of broad interest to the scientific community. I do, however, have a couple of concerns that should be addressed:

1. Standardisation of region area. SARs are not uniform among biomes, and thus the authors should account for this geographical variation. See Gerstner et al 2013 “Accounting for geographical variation in species-area relationships improves the prediction of plant species richness at the global scale”.

Response: We agree that SARs are not uniform among biomes. According to your suggestion, we have revised the standardization of area for each biome separately (lines 377-380):

Notably, the species–area relationship could have considerable geographical variation due to biomes³¹; thus, we modeled the species-area curve for each biome separately. For this, we extracted the biome type of each data point from the Terrestrial Ecoregions of the World (TEOW)³².

In addition, thank you for recommending the Gerstner et al. (2013) paper, which has been cited.

2. Biodiversity gaps. I understand that in such a global assessment there will always be data gaps. However, they must be discussed at the very least, or there should be some kind of assessment for data quality, or even better attempts to fill these gaps. For example, could it be that the result that the higher density of seed banks in the

Northern Hemisphere is simply an artefact of sampling bias? How can we believe this result when we look at the map of data locations (Fig 1.) and see these large empty spaces in the tropics and Global South?

Response: Thank you for your understanding of the data gap issue for global assessments. We have discussed the data gaps (lines 216-219):

Moreover, data gaps in our database are also likely to have had an effect on the global predictions, i.e. fewer data available from some continents (e.g. northern Asia and Africa) could lead to less confidence for prediction in these regions.

In addition, to address your concern about the potential artefact of sampling bias, we have compared seed bank diversity for each biome between Southern and Northern Hemisphere (see results in the table below). Results of the comparison were added on lines 386-393:

To account for the differences among biomes, we modeled these relationships for each biome separately. Further, to determine whether there was potential artefact of sampling bias, we compared seed bank diversity and density for each biome between Southern and Northern Hemisphere (Supplementary Table 5). Of the 9 comparisons for diversity, only 4 pairs are significantly different, among which 3 pairs actually have higher value in the Southern Hemisphere. For density, mean values were also not biased towards the Southern or Northern Hemisphere. These results clearly indicate that our database did not reflect an artefact of sampling bias.

Supplementary Table 5. Comparison of soil seed bank diversity and density between the Northern and Southern Hemisphere. mean.N, mean value in Northern Hemisphere; mean.S, mean value in Southern Hemisphere. NA, data are not sufficient for t-tests.

Code	Biome	mean.N	mean.S	t	p-value
Diversity					
1	Tropical & Subtropical Moist Broadleaf Forests	25.56	30.30	-1.73	0.09
2	Tropical & Subtropical Dry Broadleaf Forests	14.90	2.89	-	-
3	Tropical & Subtropical Coniferous Forests	37.38	NA	-	-
4	Temperate Broadleaf & Mixed Forests	20.82	25.80	-2.39	0.02
5	Temperate Conifer Forests	21.82	NA	-	-
6	Boreal Forests/Taiga	19.04	NA	-	-
7	Tropical & Subtropical Grasslands, Savannas & Shrublands	17.89	20.64	-1.00	0.32

8	Temperate Grasslands, Savannas & Shrublands	17.26	23.16	-2.80	0.01
9	Flooded Grasslands & Savannas	21.00	14.15	0.92	0.36
10	Montane Grasslands & Shrublands	18.94	17.85	0.43	0.67
11	Tundra	12.11	NA	-	-
12	Mediterranean Forests, Woodlands & Scrub	27.64	23.01	1.98	0.05
13	Deserts & Xeric Shrublands	13.08	19.50	-3.54	0.001
14	Mangroves	20.78	8.10	0.90	0.46
	Density				
1	Tropical & Subtropical Moist Broadleaf Forests	14438.04	3801.48	5.19	<0.001
2	Tropical & Subtropical Dry Broadleaf Forests	9188.91	5.02	-	-
3	Tropical & Subtropical Coniferous Forests	6014.91	NA	-	-
4	Temperate Broadleaf & Mixed Forests	15380.31	12475.1	1.08	0.28
5	Temperate Conifer Forests	247063.2	NA	-	-
6	Boreal Forests/Taiga	3768.3	NA	-	-
7	Tropical & Subtropical Grasslands, Savannas & Shrublands	2240.3	6403.7	-2.55	0.01
8	Temperate Grasslands, Savannas & Shrublands	6362.75	11518.6	-3.5	0
9	Flooded Grasslands & Savannas	1640.64	1148.79	0.68	0.5
10	Montane Grasslands & Shrublands	4241.32	1523.41	4.7	0
11	Tundra	2672.2	NA	-	-
12	Mediterranean Forests, Woodlands & Scrub	9786.77	7224.9	2.23	0.03
13	Deserts & Xeric Shrublands	14094.41	7725.67	2.42	0.02
14	Mangroves	3793.34	1028.66	1.45	0.28

We assessed seed bank diversity and density at the global scale, based on the available data that reflected how many studies have been conducted in different parts of the world. The data presented in Fig. 1 were used for finding relationships between soil seed bank and environmental variables, from which the extrapolated map was drawn. Therefore, the extrapolated map did not rely only on the data points on a particular region.

3. I could have read over it but to me it is not fully clear how you go from Fig 1. to Fig 4. If I look at Russia, which has hardly any soil seed bank data available according to Fig 1, yet then has quite a high density of seeds according to Fig 4, I get confused. Please make this clearer in text.

Response: Fig. 1 shows the location of studies from which data were extracted, then the data from these locations/studies were used to model the relationships between seed bank and environmental variables. After final models were constructed and validated, we extrapolated global maps (Fig. 4). In addition, we agree that the extremely limited data availability from Russia could lead to some inaccuracy in the mapping, and we have discussed this shortcoming on lines 218-220:

For example, Russia has very few soil seed bank data, which may have led to an inaccurate prediction for this country.

Thanks again for the interesting study and I hope the comments help.

Warm regards,

Dr. Amanda Taylor

Reviewer comments, second round -

Reviewer #1 (Remarks to the Author):

I think that the Authors responded to all my questions and comments, and also to those raised by the other Reviewers. The manuscript has improved further and I think it is a well-written, interesting and clearly important contribution to the ecological literature. Congratulations to the Authors for this great work!

Kind Regards,
Orsolya Valkó

Reviewer #2 (Remarks to the Author):

The authors have addressed most of my previous comments. I still regret that the authors do not much elaborate precise predictions on the likely effects of the tested co-variables, nor discuss their findings in the light of putative processes driving the observed relationships. But I guess that this is more of a matter of taste. I still have two remarks:

-l.237-238: "global environmental change will therefore have deleterious impact on soil seed banks"
-> erase this sentence, your study does not demonstrate that global environmental change will have deleterious impacts on soil seed banks.

-l.390-393: "Of the 9 comparisons for diversity, only 4 pairs are significantly different (...) These results clearly indicate that our database did not reflect an artefact of sampling bias" -> My understanding is quite different: the 4 significantly different pairs do pose a problem since there should not be any significant difference, or do I miss something?

Reviewer #3 (Remarks to the Author):

The Authors have made a comprehensive effort at responding to reviewer comments and as a result the manuscript is much improved. In particular, the authors discuss the data shortcomings and standardised area relative to each biome, which were my main criticisms. As before, I think this is an important synthesis and I commend the authors again on their effort to mobilise such an initiative.

Best,
Amanda Taylor

The original comments of the reviewers in black, followed by our responses in blue.

REVIEWERS' COMMENTS

Reviewer #1 (Remarks to the Author):

I think that the Authors responded to all my questions and comments, and also to those raised by the other Reviewers. The manuscript has improved further and I think it is a well-written, interesting and clearly important contribution to the ecological literature. Congratulations to the Authors for this great work!

Kind Regards,

Orsolya Valkó

Response: We appreciate the Reviewer's positive remarks on our revision and the contribution of our work.

Reviewer #2 (Remarks to the Author):

The authors have addressed most of my previous comments. I still regret that the authors do not much elaborate precise predictions on the likely effects of the tested co-variables, nor discuss their findings in the light of putative processes driving the observed relationships. But I guess that this is more of a matter of taste. I still have two remarks:

Response: We thank the Reviewer for the overall favorable comments on our revision. Also, we appreciate the Reviewer's kind understanding of the matter of taste.

-1.237-238: "global environmental change will therefore have deleterious impact on soil seed banks" -> erase this sentence, your study does not demonstrate that global environmental change will have deleterious impacts on soil seed banks.

Response: We agree and have deleted this sentence (lines 223-224).

-1.390-393: "Of the 9 comparisons for diversity, only 4 pairs are significantly different (...) These results clearly indicate that our database did not reflect an artefact of sampling bias" -> My understanding is quite different: the 4 significantly different pairs do pose a problem since there should not be any significant difference, or do I miss something?

Response: These comparisons were intended to determine whether the higher seed bank extrapolated in the Northern Hemisphere (Fig. 4) was resulted from potential artefact of sampling bias of the more sampling points in the Northern Hemisphere than in the Southern Hemisphere. The results showed that, of the 9 comparisons for diversity, only 4 pairs are significantly different, among which 3 pairs actually have

higher value in the Southern Hemisphere, which indicating that our global prediction is unlikely to reflect an artefact of sampling bias between the Northern and Southern Hemispheres. To account for the comment of the Reviewer, we have revised this statement to be more specific (lines 306-309):

These results clearly indicate that our global predictions of the higher soil seed banks in the Northern Hemisphere (Fig. 4) are unlikely to reflect an artefact of sampling bias between the Northern and Southern Hemispheres.

Reviewer #3 (Remarks to the Author):

The Authors have made a comprehensive effort at responding to reviewer comments and as a result the manuscript is much improved. In particular, the authors discuss the data shortcomings and standardised area relative to each biome, which were my main criticisms. As before, I think this is an important synthesis and I commend the authors again on their effort to mobilise such an initiative.

Best,

Amanda Taylor

Response: We thank the Reviewer for kind comments on the revision and our synthesis.